# Epigenetic age acceleration and clinical outcomes in gliomas

**Chunlei Zheng**[1], **Nathan A. Berger**[2,3], **Li Li**[4], **Rong Xu**[1,3]*

**1** Center for Artificial Intelligence in Drug Discovery, School of Medicine, Case Western Reserve University, Cleveland, OH, United States of America, **2** Center for Science, Health, and Society, School of Medicine, Case Western Reserve University, Cleveland, OH, United States of America, **3** Case Comprehensive Cancer Center, Case Western Reserve University, Cleveland, OH, United States of America, **4** Department of Family Medicine, School of Medicine, University of Virginia, Charlottesville, VA, United States of America

* rxx@case.edu

**Data Availability Statement:** All DNA methylation data are available from GDC data portal (https://portal.gdc.cancer.gov).

**Funding:** Funding for our research projects was provided by NIH (nih.gov) grants to RX, including DP2HD084068, R01 AG057557-01, R01

## Abstract

Epigenetic age acceleration—the difference between an individual's DNA methylation age and chronological age—is associated with many diseases including cancer. This study aims to evaluate epigenetic age acceleration as a prognostic biomarker for gliomas. DNA methylation data of gliomas patients (516 low-grade and intermediate-grade gliomas and 140 glioblastoma) were obtained from The Cancer Genome Atlas (TCGA) and patient epigenetic ages were computed using Horvath's age prediction model. We used multivariate linear regression to assess the association of epigenetic age acceleration with tumor molecular subtypes, including Codel, Classic-like, G-CIMP-high, G-CIMP-low, Mesenchymal-like and PA-like. Compared with Codel subtype, epigenetic ages in other molecular subtypes show deceleration after controlling age and race. Age deceleration for Classic-like, G-CIMP-high, G-CIMP-low, Mesenchymal-like and PA-like were 15.42 years (CI: 7.98–22.86, p = 5.38E-05), 25.00 years (CI: 20.79–29.22, p = 4.06E-28), 28.56 years (CI: 14.37–42.74, p = 8.75E-05), 45.34 years (CI: 38.80–51.88, p = 2.15E-36), and 53.58 years (CI: 44.90–62.26, p = 4.81E-30), respectively. Then, Cox proportional hazards regression was used to assess the association of epigenetic age acceleration with patient overall survival. Our results show epigenetic age acceleration is positively associated with patient overall survival (per 10-year age acceleration, HR = 0.89; 95%CI: 0.82–0.97; p = 9.04E-03) in multivariate analysis. When stratified by molecular subtypes, epigenetic age acceleration remains positively associated with patient survival after adjusting age and tumor grade. In conclusion, epigenetic age acceleration is significantly associated with molecular subtypes and patient overall survival in gliomas, indication that epigenetic age acceleration has potential as a quantitative prognostic biomarker for gliomas.

## Introduction

Age is a strong predictor for many diseases including cancer. Aging is accompanied by cellular and molecular changes, including genetic and epigenetic alterations of genome. Several DNA

AG061388-01 and R56 AG062272-01, and American Cancer Society (cancer.org) grant (RSG-16-049-01-MPC to R. Xu). The funders had no role in study design, data collection and analysis, decision to publish, or preparation of the manuscript.

**Competing interests:** The authors have declared that no competing interests exist.

methylation-based clocks have been developed and shown to be powerful predictors of age [1–5]. In particular, Horvath's clock, a multiple tissue age predictor based on methylation of 353 CpG loci, is robustly correlated with chronological age [4]. Epigenetic age acceleration–the difference between epigenetic age and chronological age—has been shown associated with many diseases, including Parkinson's disease [6], Down syndrome [7], obesity [8], Alzheimer's disease [9], and others. In addition, epigenetic age acceleration shows predictive power for morbidity and mortality [10–13]. Though epigenetic age acceleration was observed in many cancers [4], it remains unclear whether epigenetic age can be used as biomarkers for cancer prognosis.

Gliomas are the most prevalent primary brain tumors, among which diffuse low-grade and intermediate-grade gliomas (LGG) and grade IV glioblastomas (GBM) are the major groups [14]. The five-year survival rate for LGG is high (more than 75%), while the overall survival for GBM is only 15 months [14, 15]. A number of molecular biomarkers for gliomas have been identified. For example, mutations in IDH1 is associated with better survival in younger patients of GBM [16]; 1p/19q deletion is strongly associated with oligodendroglia differentiation and better response to chemical therapies in oligodendroglioma patients [17]. These molecular biomarkers have a significant impact on the diagnosis and management of gliomas [18–20]. However, these single gene-based biomarkers are qualitative and often only present in a subset of specific types of cancer patients. For example, mutations in IDH1 only occur in 12% of GBM patients [16] and 1p/19q deletion are mostly found in oigodendrogliomas, but not GBM [18]. Therefore, there is a need to develop quantitative biomarkers that are also applicable in multiple types of gliomas. Here we evaluate the potential of epigenetic age acceleration as quantitative prognostic biomarkers for broad types of gliomas, including both GBM and LGG.

A recent study suggested that epigenetic aging can serve as a potential prognostic biomarker for gliomas and showed that epigenetic age was correlated with molecular subtype of gliomas and significantly associated with patient survival [21]. Since epigenetic age is highly correlated with chronological age, which is a strong predictor for cancer patient survival, the previous study of the role of epigenetic age in predicting patient survival in glioma patients has the inherent limitation in separating the contributions of chronological age from that of epigenetic aging for patient outcome. In this study, we aim to evaluate the potential of epigenetic age acceleration (epigenetic aging after controlling the effects of biological aging) as a prognostic biomarker for gliomas.

The association of epigenetic age acceleration with cancer patient outcomes has been investigated in several studies. A pan-cancer study using DNA methylation data from The Cancer Genome Atlas (TCGA) reported that the association of epigenetic age acceleration with patient survival varies with cancer types [22]. Positive association was observed for esophageal carcinoma while negative association was observed for both thyroid carcinoma and renal clear cell carcinoma. No significant association was observed for lung adenocarcinoma, colon adenocarcinoma, pancreatic adenocarcinoma and GBM. One limitation of this pan-cancer study is that the models only adjusted chronological age. A recent study of breast cancer reported that clinical and molecular features such as molecular subtypes and tumor grade, are associated with both patient survival and epigenetic age acceleration [23], suggesting molecular and clinical features in addition to chronological age should be considered in the analysis of associations between epigenetic age acceleration and patient survival. Another study using data nested in the Melbourne Collaborative Cohort Study assessed the associations of epigenetic age acceleration with cancer risk and survival for seven common cancers [24] and found no association of epigenetic age acceleration with patient survival after adjusting sociodemographic and lifestyle variables. This previous study was based on epigenetic age data from blood samples of cancer

patients due to the unavailability of the tumor tissue samples, however epigenetic age acceleration in tumor tissue samples are often different from that in blood samples. In this study, we examined the association of epigenetic age acceleration and clinical outcomes of gliomas using tumor tissue samples.

We recently reported that epigenetic age acceleration is significantly associated with consensus molecular subtypes (CMS) of colorectal cancer in the TCGA patients [25]. Compared with CMS2, epigenetic age acceleration for CMS1, CMS3, and CMS4 was 23.90 years, 9.16 years, and 6.05 years, respectively. Furthermore, epigenetic age acceleration is positively associated with total mortality (HR = 1.97; 95%CI: 1.14–3.39; $P$ = 0.014). Our previous study demonstrated the importance of incorporating molecular subtype in the analysis of the association of epigenetic age acceleration with patient survival. Leveraging multiple robust molecular subtyping platforms and extensive DNA methylation data for gliomas available in TCGA, we here evaluated epigenetic age acceleration as a potential biomarker for glioma patient survival, with an emphasis on its association with molecular subtypes and tumor grade. We found that accelerated epigenetic age is significantly associated with better patient survival of gliomas, which is opposite to that observed in colorectal cancer [25].

## Materials and methods

### Study population

A total of 516 LGG and 140 GBM patients from TCGA were included in this study. DNA methylation data (Illumina 450K platform) of brain tissue and clinic information of these patients were downloaded from TCGA. Patient characteristics was shown in S1 Table. We exclude 41 patients from this original data, including 4 patients who don't have age information, 9 patients from minorities (one is Native and 8 are Asian) due to small sample size in these two groups and 28 patients who don't have molecular subtype information, which led to 615 patients for analyses. There are multiple molecular subtype classification platforms available in TCGA. In this study, we used the Supervised DNA methylation (SDM) system based on its better clinical relevance [26, 27], which classified brain tumors patients into six types: Codel (IDH mutant-codel LGGs), G-CIMP-high (IDH mutant-non-codel glioma with higher global levels of DNA methylation), G-CIMP-low (IDH mutant-non-codel glioma with relatively low genome-wide DNA methylation), Classic-like (IDH wild type with classical gene expression signature) and Mesenchymal-like and PA-like (pilocytic astrocytoma).

In addition, we compiled a validation dataset from three published studies under Gene Expression Omnibus resources (GSE36278, GSE61160 and GSE44684) [28–30]. These GEO series including 136, 32 and 61 glioma patients respectively. We'd like to mention another dataset GSE30338 that includes 81 glioma patients and was used in another study [21]. However, DNA methylation data in that dataset were unreversed transformed and the raw β values are not available, which is required for computing epigenetic age. Hence, we didn't include this dataset in our study. The SDM subtypes were obtained from a supervised random forest model [31]. Patient statistics was shown in S2 Table.

### DNA methylation age and epigenetic age acceleration

We used Horvath's model to calculate DNA methylation age [4]. The Horvath's model uses beta values of 353 CpG loci to calculate DNA methylation age as following:

$$DNAmAge = inverse.F(b_0 + b_1 CpG_1 + \cdots + b_{353} CpG_{353})$$

where $F$ is a function for transformation of age and $b_0, b_1 \ldots b_{353}$ are coefficients obtained from

the elastic net regression model. Epigenetic age acceleration is then estimated as the residual of regression of DNA methylation age on chronological age [24, 25].

## Statistical analysis

All statistical analyses were performed using R (Version: 3.5.2). Multivariable linear regression model was used to assess the association of epigenetic age acceleration with tumor molecular subtype and tumor grade. Kaplan-Meier curves were used to estimate survival rates of patients with different molecular subtypes and tumor grades, and log-rank test was used to test the significance of difference. Cox proportional hazards regression was used to assess the association of epigenetic age acceleration with patient overall survival in both stratified and un-stratified analyses.

## Results

### Epigenetic age acceleration is associated with DNA methylation-based subtypes [SDM]

This study analyzes existing individual patient data from TCGA, which overrepresents white patients compared with the US population and underrepresents primarily Asian and Hispanic patients [32]. The glioma patients in TCGA for this study were mostly white (90.4%). Tumor histology and gender were evenly distributed. Over one hundred patients were included in each tumor grade. All six molecular subtypes of gliomas, including Codel (IDH mutant-codel LGGs), G-CIMP-high (IDH mutant-non-codel glioma with higher global levels of DNA methylation), G-CIMP-low (IDH mutant-non-codel glioma with relatively low genome-wide DNA methylation), Classic-like (IDH wild type with classical gene expression signature) and Mesenchymal-like and PA-like (pilocytic astrocytoma), were represented in this dataset (Table 1).

Based on univariate analysis, epigenetic age acceleration was significantly associated with race, histology type, tumor grade and SDM subtypes, but not with gender (Table 1). In addition, epigenetic age acceleration shows suggestive, but not significant association with two chronological age groups ($< 60$ years and $> 60$ years with p = 0.068). We used multivariate linear regression to assess whether epigenetic age acceleration is independently associated with SDM. Due to high correlation of histology and tumor grade, histology was not included in the covariates. After adjusting age group, race and tumor grade, epigenetic age acceleration remains significantly associated with molecular subtypes. Compared to Codel subtype, the other molecular subtypes show age deceleration ranging from 15.42 (CI: -22.86 - -7.98, p = 5.38E-05) for Classic-like to 53.58 (CI: -62.26 - -44.90, p = 4.81E-30) for PA-like (Table 2).

We then validated this association in the compiled validation dataset including 229 glioma patients. Both in univariate and multivariate regression analysis, we see the significant association of epigenetic age acceleration with molecular subtype (S2 and S3 Tables). Consistent with the results from discovery dataset, epigenetic age shows deceleration in other molecular subtypes compared to Codel subtype. Furthermore, the age deceleration showed bigger in the order of Class-like, G-CIMP-high, G-CIMP-low, Mesenchymal-like and PA-like, which is also concordant with that in the discovery dataset. In summary, the association of epigenetic age acceleration with molecular subtype is independently validated.

### Epigenetic age acceleration is positively associated with patient overall survival in univariate analysis

We used two methods to investigate the relationship of epigenetic age acceleration with patient overall survival in gliomas: Kaplan-Meier estimator and Cox proportional hazards regression.

**Table 1. Patient characteristics and the associations of epigenetic age acceleration with clinical variables.**

|  | Patient (%) | Epigenetic age acceleration Mean (years) | p[a] |
|---|---|---|---|
| **Age** |  |  |  |
| < 60 years | 489 (79.5) | 1.84 | 0.068 |
| > 60 years | 126 (20.5) | -3.64 |  |
| **Gender** |  |  |  |
| female | 272 (44.2) | 1.69 | 0.405 |
| male | 343 (55.8) | -0.06 |  |
| **Race** |  |  |  |
| white | 556 (90.4) | 1.49 | 0.036 |
| black | 43 (6.99) | -7.08 |  |
| unknown | 16 (2.6) | -5.41 |  |
| **Histology** |  |  |  |
| astrocytoma | 166 (27) | -4.62 | **1.90E-15** |
| glioblastoma | 108 (17.6) | -10.63 |  |
| oligoastrocytoma | 113 (18.4) | 0.56 |  |
| oligodendroglioma | 166 (27) | 13.12 |  |
| unknown | 62 (10.1) | 1.82 |  |
| **Tumor grade** |  |  |  |
| G2 | 208 (33.8) | 3.03 | **1.85E-06** |
| G3 | 237 (38.5) | 3.56 |  |
| G4 | 108 (17.6) | -10.63 |  |
| unknown | 62 (10.1) | 1.82 |  |
| **Molecular subtype** |  |  |  |
| Codel | 168 (27.3) | 22.92 | **2.20e-16** |
| Classic-like | 73 (11.9) | 5.7 |  |
| G-CIMP-high | 238 (38.7) | -3.48 |  |
| G-CIMP-low | 11 (1.79) | 4.44 |  |
| Mesenchymal-like | 100 (16.3) | -23.02 |  |
| PA-like | 25 (4.07) | -29.77 |  |

[a] For 2-level variables, t-test was used, for more than 2-level variables, one-way ANOVA test was used. Unknown data were not used in tests

We divided epigenetic age acceleration into two groups: age deceleration and age acceleration, to facilitate analyses using Kaplan-Meier estimator and investigate the overall association of epigenetic age acceleration with patient survival. Kaplan-Meier curves show that epigenetic age acceleration group has significantly better survival than age deceleration group (Fig 1). Since other clinical factors, such as tumor grade and histology type, are shown significant association with glioma patient survival (S1 Fig), we stratified patients according to their tumor grade and histology type and performed survival analysis in stratified patient population. We show that patients with epigenetic age acceleration have improved survival for both Grade 2 (S2 Fig) and oligoastrocytoma (S3 Fig).

Next, we used univariate Cox proportional hazards regression to compare the patient overall survival in each clinic groups (Table 3). Compared to Grade 2 patients, Grade 3 and 4 patients have worse survival with hazard ratios of 2.95 (95% CI: 1.92–4.53, p = 7.48E-7) and 14.7 (95% CI: 9.4–22.99, p = 4.61E-32) respectively. Using Codel subtype as the reference, other molecular subtypes show worse survival, especially for Classic-like (HR:14.71, CI: 8.69–24.87, p = 1.20E-23), mesenchymal-like (HR: 23.35, CI: 13.71–39.78, p = 4.60E-31) and G-CIMP-low (HR: 8.07, CI: 3.21–20.25, p = 8.75E-06). Since epigenetic age acceleration in

**Table 2. The associations of epigenetic age acceleration with clinical variables.**

| | Epigenetic age acceleration (years) | 95% CI (Lower) | 95% CI (Upper) | P value |
|---|---|---|---|---|
| **Molecular subtype (Codel as Ref.)** | | | | |
| Classic-like | -15.42 | -22.86 | -7.98 | **5.38E-05** |
| G-CIMP-high | -25.00 | -29.22 | -20.80 | **4.06E-28** |
| G-CIMP-low | -28.56 | -42.74 | -14.37 | **8.75E-05** |
| Mesenchymal-like | -45.34 | -51.88 | -38.80 | **2.15E-36** |
| PA-like | -53.58 | -62.26 | -44.90 | **4.81E-30** |
| **Age (<60 years as Ref.)** | | | | |
| > 60 years | 1.05 | -3.61 | 5.71 | 0.658 |
| **Race (White as Ref.)** | | | | |
| black | -3.27 | -10.38 | 3.84 | 0.367 |
| **Tumor grade (G2 as Ref.)** | | | | |
| G3 | 5.03 | 1.11 | 8.94 | **0.012** |
| G4 | 0.04 | -7.03 | 7.10 | 0.992 |

Multivariate linear regression was used to study the association of epigenetic age acceleration with SDM, adjusted by age, race, and tumor grade.

glioma patients has large range (min: -53.4, max: 89.1, median: -2.6), we scaled down epigenetic age acceleration by a factor of 10 and investigated the relationship of 10-year epigenetic age change with patient survival as in previous studies (23, 24, 25). We show that epigenetic age acceleration is significantly associated with patient survival (per 10-year age acceleration, HR = 0.86, CI: 0.80–0.91, p = 1.20E-06).

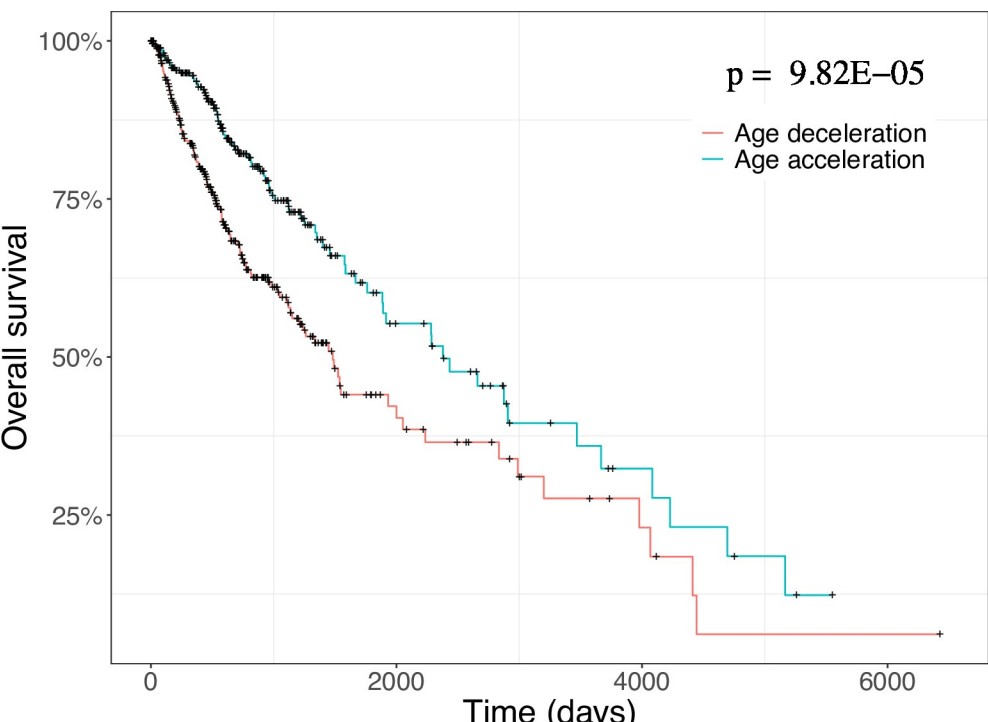

**Fig 1. Kaplan-Meier curves for patient overall survival between epigenetic age acceleration and epigenetic age deceleration.**

**Table 3. Overall survival of gliomas patients in univariate analysis.**

| | Number of patients | Death | Death rate | HR (95% CI) | P value |
|---|---|---|---|---|---|
| **Age acceleration** | 615 | 200 | 32.5 | 0.86 (0.80,0.91) | **1.20E-06** |
| **Age group** | | | | | |
| < 60 years | 489 | 122 | 24.9 | Reference | |
| > 60 years | 126 | 78 | 61.9 | 5.66 (4.2,7.63) | **5.06E-30** |
| **Gender** | | | | | |
| female | 272 | 89 | 32.7 | Reference | |
| male | 343 | 111 | 32.4 | 1.07 (0.81,1.41) | 0.654 |
| **Race** | | | | | |
| white | 556 | 178 | 32 | Reference | |
| black | 43 | 19 | 44.2 | 1.87 (1.16,3.01) | **9.75E-03** |
| unknown | 16 | 3 | 18.8 | | |
| **Histology** | | | | | |
| astrocytoma | 166 | 46 | 27.7 | Reference | |
| glioblastoma | 108 | 74 | 68.5 | 5.61 (3.84,8.2) | **5.02E-19** |
| oligoastrocytoma | 113 | 24 | 21.2 | 0.67 (0.41,1.1) | 0.116 |
| oligodendroglioma | 166 | 36 | 21.7 | 0.58 (0.37,0.9) | **1.61E-02** |
| unknown | 62 | 20 | 32.3 | | |
| **WHO grade** | | | | | |
| G2 | 208 | 31 | 14.9 | Reference | |
| G3 | 237 | 75 | 31.6 | 2.95 (1.92,4.53) | **7.48E-07** |
| G4 | 108 | 74 | 68.5 | 14.7 (9.4,22.99) | **4.61E-32** |
| unknown | 62 | 20 | 32.3 | | |
| **Molecular subtype** | | | | | |
| Codel | 168 | 22 | 13.1 | Reference | |
| Classic-like | 73 | 51 | 69.9 | 14.71 (8.69,24.87) | **1.20E-23** |
| G-CIMP-high | 238 | 47 | 19.7 | 1.42 (0.85,2.38) | 0.182 |
| G-CIMP-low | 11 | 6 | 54.5 | 8.07 (3.21,20.25) | **8.75E-06** |
| Mesenchymal-like | 100 | 68 | 68 | 23.35 (13.71,39.78) | **4.60E-31** |
| PA-like | 25 | 6 | 24 | 1.94 (0.77,4.86) | 0.159 |

Univariate Cox proportional hazards regression was used to fit the data and likelihood ratio test was used to compute the p value

### Epigenetic age acceleration is positively associated with patient overall survival in multivariate analysis

To assess whether epigenetic age acceleration can provide independently prognostic information besides survival predictors mentioned above, such as age, tumor grade, histology and molecular subtype, we performed survival analysis using multivariate Cox proportional hazards regression. Due to high correlation of histology and tumor grade, histology was not included in the covariates. Our analysis shows that epigenetic age acceleration is positively associated with patient survival (per 10-year age acceleration, HR = 0.89, CI: 0.82–0.97, p = 9.04E-03) after adjusting age, race, tumor grade and SDM molecular subtype (Table 4), indicating that epigenetic age acceleration is an independent prognostic factor for glioma patients.

### Epigenetic age acceleration is variably associated with patient overall survival in SDM subtypes

To investigate how epigenetic age acceleration is associated with overall survival in patient population-specific ways, we stratified patients by age, tumor grade and molecular subtype

**Table 4. Overall survival of gliomas patients in multivariate analysis.**

|  | HR | 95% CI (Lower) | 95% CI (Upper) | P value |
|---|---|---|---|---|
| **Age acceleration** | 0.89 | 0.82 | 0.97 | **9.04E-03** |
| **Molecular subtype (Codel as Ref.)** |  |  |  |  |
| Classic-like | 6.43 | 3.22 | 12.82 | **1.32E-07** |
| G-CIMP-high | 1.04 | 0.58 | 1.85 | 0.896 |
| G-CIMP-low | 4.04 | 1.31 | 12.41 | **1.49E-02** |
| Mesenchymal-like | 8.48 | 4.20 | 17.11 | **2.43E-09** |
| PA-like | 0.88 | 0.30 | 2.52 | 0.805 |
| **Age (<60 years as Ref.)** |  |  |  |  |
| > 60 years | 2.25 | 1.56 | 3.25 | **1.45E-05** |
| **Race (White as Ref.)** |  |  |  |  |
| black | 1.29 | 0.74 | 2.24 | 0.368 |
| **Tumor grade (G2 as Ref.)** |  |  |  |  |
| G3 | 1.81 | 1.11 | 2.94 | **1.70E-02** |
| G4 | 2.03 | 1.09 | 3.76 | **2.50E-02** |

Cox proportional hazards regression was used for multivariate survival analysis to assess the association of patient characteristic with overall survival

and assessed the association of epigenetic age acceleration with patient overall survival in each group. The epigenetic age acceleration shows similar positive associations with patient survival both in younger and older groups (Fig 2A). The epigenetic age acceleration shows similar but distinctive positive associations in each tumor grade group (Fig 2B).

When patients were stratified by SDM subtype, epigenetic age acceleration shows positive association with patient survival in Classic-like and Mesenchymal-like subtypes, but negative association with patient survival in Codel subtype. The associations in G-CIMP-high and G-CIMP-low are not significant (Fig 3A). Since age and tumor grade are two key predictors for patient survival, we evaluated the epigenetic age acceleration with glioma patient survival after adjusting age and tumor grade. We can see a left shift of hazard ratio in each molecular subtype (Fig 3B). Suggestive positive associations of epigenetic age acceleration with patient survival were observed in G-CIMP-high and G-CIMP-low subtypes.

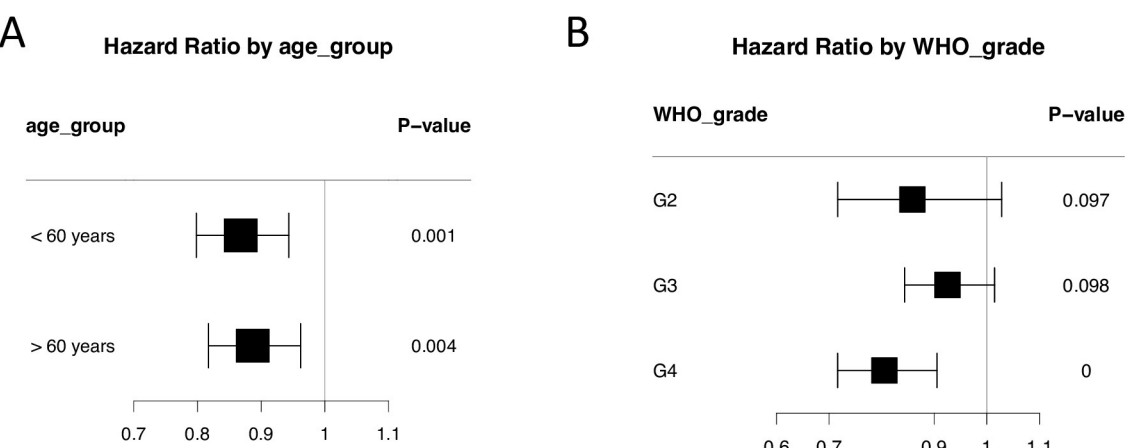

**Fig 2.** Forest plot shows associations of age acceleration with patient overall survival in stratified patient groups: (A) Age group; (B) Tumor grade.

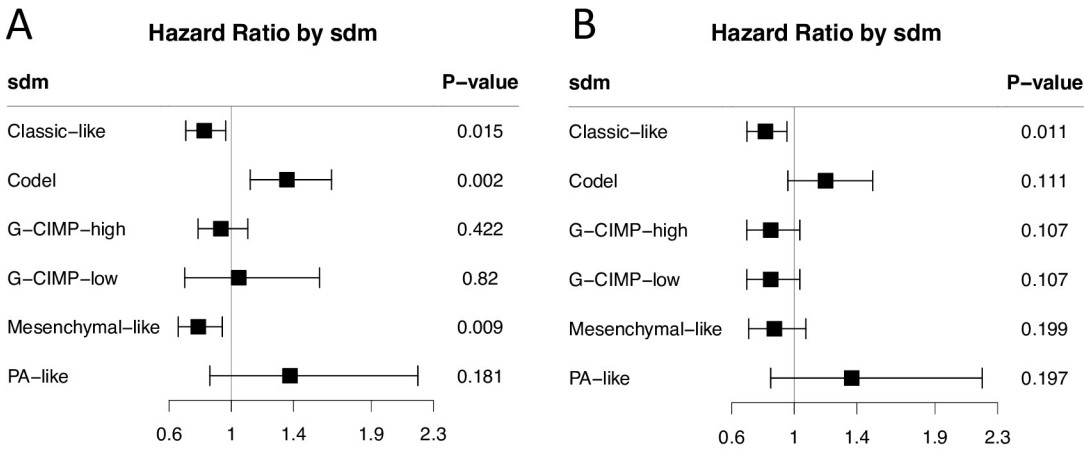

Fig 3. Forest plot shows the association of age acceleration with patient overall survival in SDM molecular subtype. (A) No adjusting. (B) Adjusting for age and tumor grade.

**Validation of the association of epigenetic age acceleration with patient overall survival.** We attempted to perform similar survival analyses in the validation dataset. Since limited survival information is available, a total of 105 of 229 glioma patients are eligible for survival analysis. After adjusting age, grade and molecular subtypes, we didn't see the significant association of age acceleration with patient overall survival (S4 Table). Unable to validate the association of epigenetic age acceleration with patient overall survival is likely due to limitation of validation data. First, the compiled validation data were from multiple studies, which may lead to its bias for particularly population or more heterogenous. For example, validation dataset has higher percentage of younger patient (93.4%). Heterogeneity of the population from multiple studies makes it difficult to control the confounders. Second, the sample size is small. The validation dataset has just 105 samples, which will have limited power to detect difference. In fact, we see large variation of hazard ratios for each variable (S4 Table). Hence, we hypothesize that the heterogeneous population with small sample size may account for the failure to replicate the survival results.

We then performed survival analyses in stratified patients according to age, tumor grade and molecular subtype. Interestingly, we observed the trends that epigenetic age acceleration has benefit for glioma patients similar to the findings in the discover dataset. In the younger patients, the hazard ratio is 0.93 (95% CI: 0.84–1.03, p = 0.152). Similarly, the hazard ratios in G2, G3 and G4 are 0.77 (95% CI: 0.47–1.28, p = 0.317), 0.87 (95% CI: 0.71–1.11, p = 0.299) and 0.97 (95% CI: 0.85–1.10, p = 0.640) respectively For the hazard ratios in different molecular subtypes, epigenetic age acceleration shows marginally positive association with patient survival in Mesenchymal-like subtype (HR: 0.85, 95% CI: 0.70–1.13, p = 0.094) (S5 Table and Fig 4).

In summary, although we are unable to completely validate the positive association of epigenetic age acceleration with patient overall survival due to heterogeneity and small sample size of validation data, we observed consistent results in both discovery and validation data in the stratified analyses.

## Discussion

In this study, we evaluated the associations of epigenetic age acceleration with patient survival in gliomas. Our results show that epigenetic age acceleration is significantly associated with

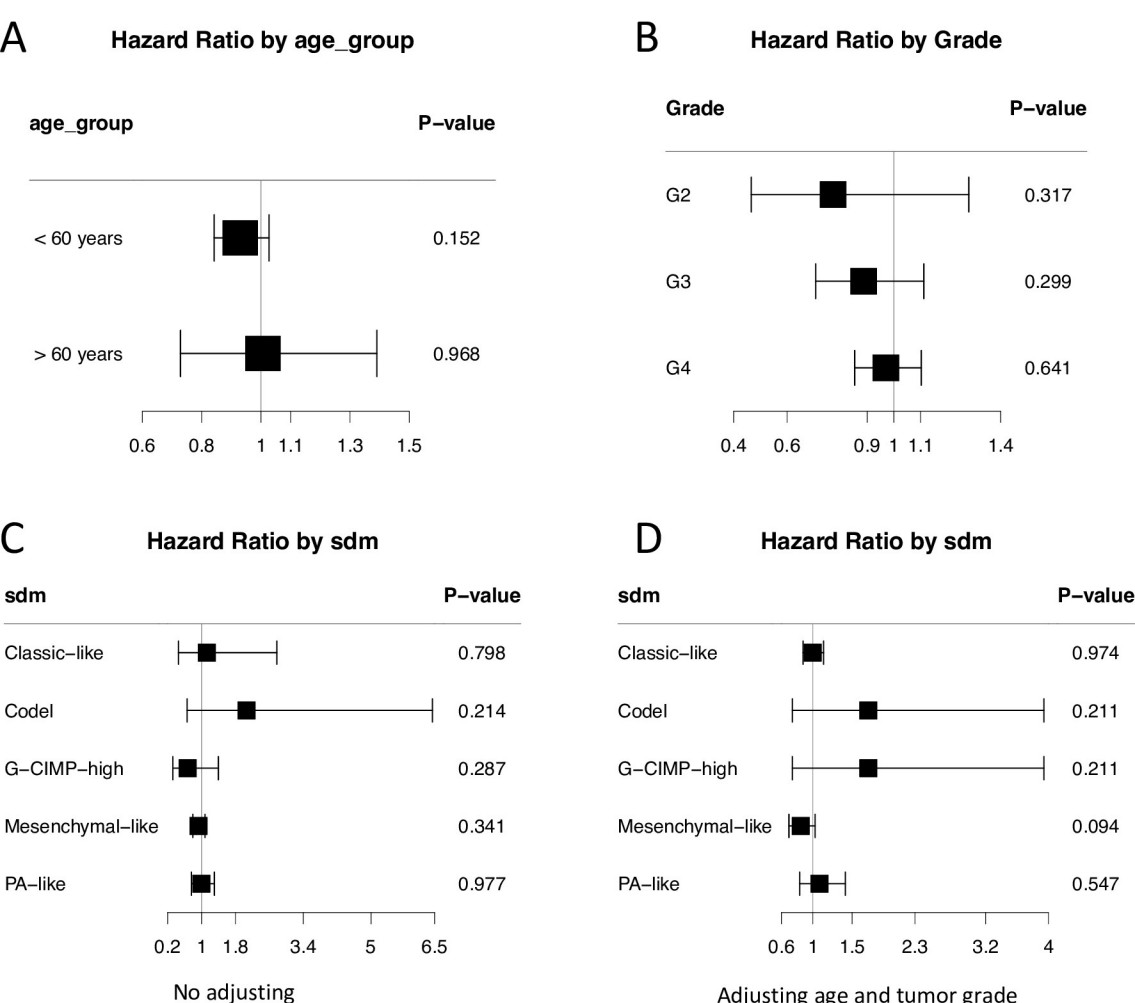

**Fig 4. Forest plot shows the association of age acceleration with patient overall survival in each subgroup.** (A) Age group. (B) Tumor grade. (C) Molecular subtype without adjusting. (D) Molecular subtype adjusting for age and tumor grade.

DNA methylation-based molecular subtypes in gliomas. By incorporation of molecular subtype in our survival analysis, epigenetic age acceleration was shown to be significantly associated with patient overall survival. When we stratified patients based on molecular subtypes, this significant association still exists in 4 of 6 molecular subtypes after adjusting age and tumor grade. Taken together, the evidence suggests that epigenetic age acceleration has potential as promising prognostic biomarker for glioma patients. However, our findings, though statistically significant, should be interpreted cautiously because of small sample size (516 low-grade and intermediate-grade gliomas and 140 glioblastoma).

Several molecular biomarkers for gliomas have been identified in recent decades, including IDH1 mutation [16], 1p/19q deletion [17], MGCT promoter methylation [33] and EGFRvIII [34], among others. These molecular biomarkers have made great contributions in diagnosis, therapeutic decision and prognosis of gliomas. However, these single gene-based biomarkers are often present in a subset of specific types of cancer patients. In this study, we show that epigenetic age acceleration is significantly associated with patient overall survival in gliomas. Compared to traditional single gene-based molecular biomarkers for a subset of cancer patients, epigenetic age acceleration is computed based on methylation of 353 CpG sites as a

summary measurement for DNA methylation, therefore biomarkers based on epigenetic age acceleration have potential as a more comprehensive predictors of clinical outcomes in a larger population of patients. In addition, biomarkers based on epigenetic age acceleration is quantitative, which can provide precise and fine-grained prediction for patient survival. As shown in our study, the risk of death is 19.5% lower for every 10-year age acceleration in Grade IV glioma patients, and 8.5% in Grade III patients (Fig 2B).

The association of epigenetic age acceleration with patient survival in gliomas is strong and independent (per 10-year age acceleration, HR = 0.89; 95%CI: 0.82–0.97; p = 9.04E-03), further supporting its potential as a prognostic biomarker in clinical settings. However, the strength of the association is cancer-specific. In our previous study, we showed a weak association of epigenetic age acceleration with patient survival in colorectal cancer, where the association is only observed when categorizing patients into epigenetic age acceleration and epigenetic age deceleration groups [25]. The mechanisms underlying this observed cancer-type specific association between epigenetic age acceleration and patient survival remains unknown and warrant further investigation. One explanation is that epigenetic age in specific cancer types is correlated with cancer-specific gene mutations. Thus, in gliomas, the epigenetic age may be associated with gene mutations that have strong predictive power for patient survival.

The direction of the association between epigenetic age acceleration and cancer patient survival is also cancer-type specific. In this study, we showed that epigenetic age acceleration is positively associated with patient survival in gliomas, suggesting cancer patients with older epigenetic age have better survival. On the other hand, negative associations of epigenetic age acceleration with patient survival were observed in other cancers, including colorectal cancer in our previous study [25] and thyroid carcinoma [22]. It remains unclear what causes this intriguing cancer-type specific relationship between epigenetic age acceleration and patient survival. Compared to normal tissues where epigenetic age is highly correlated with chronological age, cancer tissue often has disrupted epigenetic age clock that does not necessarily reflect chronological age. Age-related CpGs, especially hypermethylated CpGs, are coordinately regulated in cancer [22] and strongly enriched in CpG islands and enhancer-related loci [34–38]. Therefore, age-related CpGs likely interact with promoters to regulate target gene expression or alter genome stability to introduce gene mutations. Horvath and Lin observed that epigenetic age is often associated with mutation patterns in cancer [4, 22]. For example, mutations in TP53 have higher incidence with younger epigenetic age [22]. High prevalence of TP53 mutations was found in gliomas [39] and correlated with worse prognosis [40], which may partially explain younger epigenetic age is detrimental to glioma patients. Overall, the observed cancer-type specific association of epigenetic age acceleration with patient survival is likely related to underlying disease mechanisms in cancer-specific ways.

## Supporting information

**S1 Fig. Kaplan-Meier curves for patient overall survival in different clinical groups.**
(DOCX)

**S2 Fig. Kaplan-Meier curves for patient overall survival between epigenetic age acceleration and epigenetic age deceleration in different tumor grades.**
(DOCX)

**S3 Fig. Kaplan-Meier curves for patient overall survival between epigenetic age acceleration and epigenetic age deceleration in different histology subtypes.**
(DOCX)

**S1 Table. Patient characteristics of original data from TCGA.**
(DOCX)

**S2 Table. Patient characteristics of validation data.**
(DOCX)

**S3 Table. The associations of epigenetic age acceleration with clinical variables in validation data.**
(DOCX)

**S4 Table. Overall survival of gliomas patients in validation dataset using multivariate analysis.**
(DOCX)

**S5 Table. The association of age acceleration with patient overall survival in stratified analyses.**
(DOCX)

## Author Contributions

**Conceptualization:** Chunlei Zheng, Rong Xu.

**Formal analysis:** Chunlei Zheng.

**Funding acquisition:** Rong Xu.

**Investigation:** Chunlei Zheng, Rong Xu.

**Methodology:** Chunlei Zheng.

**Resources:** Rong Xu.

**Supervision:** Rong Xu.

**Visualization:** Chunlei Zheng.

**Writing – original draft:** Chunlei Zheng.

**Writing – review & editing:** Nathan A. Berger, Li Li, Rong Xu.

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
