## [Decision Letter · Decision Letter 0]

8 May 2020

PONE-D-20-06616

Epigenetic age acceleration and clinical outcomes in gliomas

PLOS ONE

Dear Dr Xu,

Thank you for submitting your manuscript to PLOS ONE. After careful consideration, we feel that it has merit but does not fully meet PLOS ONE’s publication criteria as it currently stands. Therefore, we invite you to submit a revised version of the manuscript that addresses the points raised during the review process.

This study is well performed and the results are novel and of potential interest to the readers. However, there are several points which need to be addressed before acceptance. Please respond to each of the reviewer comments.

We would appreciate receiving your revised manuscript by Jun 22 2020 11:59PM. To enhance the reproducibility of your results, we recommend that if applicable you deposit your laboratory protocols in protocols.io, where a protocol can be assigned its own identifier (DOI) such that it can be cited independently in the future. For instructions see: http://journals.plos.org/plosone/s/submission-guidelines#loc-laboratory-protocols

We look forward to receiving your revised manuscript.

Kind regards,

Hiromu Suzuki, M.D., Ph.D.

Academic Editor

PLOS ONE

Journal Requirements:

Reviewers' comments:

Reviewer's Responses to Questions

**Comments to the Author**

1. Is the manuscript technically sound, and do the data support the conclusions?

Reviewer #1: Yes

Reviewer #2: Yes

2. Has the statistical analysis been performed appropriately and rigorously? 

Reviewer #1: Yes

Reviewer #2: Yes

3. Have the authors made all data underlying the findings in their manuscript fully available?

Reviewer #1: Yes

Reviewer #2: No

4. Is the manuscript presented in an intelligible fashion and written in standard English?

Reviewer #1: Yes

Reviewer #2: Yes

5. Review Comments to the Author

Reviewer #1: Rong Xu et al. reported that epigenetic age acceleration is significantly associated with molecular subtypes and overall survival in glioma patients, indicating of epigenetic age acceleration has potential as a quantitative prognostic biomarker for gliomas. The authors conducted analysis well, and this manuscript is well written. It may be much more interesting for general readers of PLoS One to show what kinds of gene group is associated to epigenetic age acceleration.

Minor point.

1. In Table 1: p value of Race is missing.

Reviewer #2: The present study analyzing epigenetic age acceleration in glioma and conclude that epigenetic age acceleration is

significantly associated with molecular subtypes and patient overall survival in gliomas, indication that epigenetic age acceleration has potential as an independent and strong prognostic biomarker for patients with glioma.

Although some notions has been implied in the previous study "Models of epigenetic age capture patterns of DNA methylation in glioma associated with molecular subtype, survival, and recurrence, Neuro-Oncology, Volume 20, Issue 7, July 2018 (ref.21)", this study provide more comprehensive investigation.

Only few concerns need to address before publication:

1. Compared the ref.21 and this manuscript, similar glioma data from TCGA could be used (516 LGG and 141 GBM), but the number of histological type and WHO grade are very different. Can authors explain why, and provide the exact cohort that used in this study in Supporting information.

2. Can authors analyze any other independent glioma data set to support or show correlation to the conclusion of this study?

6. PLOS authors have the option to publish the peer review history of their article (what does this mean?). If published, this will include your full peer review and any attached files.

Reviewer #1: No

Reviewer #2: No

---

## [Author Response · Author response to Decision Letter 0]

21 Jun 2020

Reviewer #1: Rong Xu et al. reported that epigenetic age acceleration is significantly associated with molecular subtypes and overall survival in glioma patients, indicating of epigenetic age acceleration has potential as a quantitative prognostic biomarker for gliomas. The authors conducted analysis well, and this manuscript is well written. It may be much more interesting for general readers of PLoS One to show what kinds of gene group is associated to epigenetic age acceleration.

Epigenetic age, especially Horvath’s clock here, is defined as a function of DNA methylation of 353 CpG Sites. In the original paper (see below for reference), Dr. Horvath proposed that epigenetic age models “Epigenetic Maintenance System”. Among 353 CpGs, 193 CpGs are positively correlated and 160 are negatively correlated with age. Different from epigenetic age models where CpGs can be mapped to their nearest genes, epigenetic age acceleration is defined as the difference between epigenetic age and chorological age. Due to the inclusion of chronological age in the model, genes associated with epigenetic age acceleration can no longer be directly mapped from CpGs as did for epigenetic age models. It remains an open question to understand the molecular mechanisms underlying epigenetic age acceleration and how it correlates to patient disease characteristics and outcomes 

Horvath S. DNA methylation age of human tissues and cell types. Genome Biol. 2013;14(10): R115.

Minor point.

1. In Table 1: p value of Race is missing.

We added the p value for Race and corrected the p values for other variables accordingly in this revision.

Reviewer #2: The present study analyzing epigenetic age acceleration in glioma and conclude that epigenetic age acceleration is significantly associated with molecular subtypes and patient overall survival in gliomas, indication that epigenetic age acceleration has potential as an independent and strong prognostic biomarker for patients with glioma.

Although some notions has been implied in the previous study "Models of epigenetic age capture patterns of DNA methylation in glioma associated with molecular subtype, survival, and recurrence, Neuro-Oncology, Volume 20, Issue 7, July 2018 (ref.21)", this study provide more comprehensive investigation.

Only few concerns need to address before publication:

1. Compared the ref.21 and this manuscript, similar glioma data from TCGA could be used (516 LGG and 141 GBM), but the number of histological type and WHO grade are very different. Can authors explain why, and provide the exact cohort that used in this study in Supporting information.

Thanks for this point. We used the same dataset as in ref.21. Different from ref.21 study, we focused on investigating the association between age acceleration and molecular subtypes. Therefore, we excluded from the original dataset 28 patients without molecular subtype information, 4 patients without age information, 9 patients from minorities (one is Native and 8 are Asian) due to small sample size in these two groups,

We clarified the data processing procedure in “Study population” of “Materials and Methods” section and provided the detailed original information in S1 Table. 

2. Can authors analyze any other independent glioma data set to support or show correlation to the conclusion of this study?

To validate our results, we compiled a validation cohort from three published studies including 229 patients under GEO (GSE36278, GSE61160 and GSE44684, see S2 Table for the details). Using similar analyses, we pursued to replicate two key findings in this study.

1) The association of epigenetic age acceleration with molecular subtype.

We can clearly see that epigenetic age acceleration is significantly associated with molecular subtype, which is consistent with the result from discover dataset (See S3 Table). 

S3 Table The associations of epigenetic age acceleration with clinical variables in validation data a

 Epigenetic age 

acceleration (years) 95% CI 

(Lower) 95% CI 

(Higher) P value

Molecular subtype (Codel as Ref.)

 Classic-like -16.49 -30.62 -2.37 2.23E-02 *

 G-CIMP-high -26.94 -40.72 -13.16 1.54E-04 ***

 G-CIMP-low -28.26 -46.52 -10.01 2.56E-03 **

 Mesenchymal-like -34.28 -46.49 -22.06 9.13E-08 ***

 PA-like -44.43 -57.65 -31.22 2.63E-10 ***

Age (<=60 as Ref.)

 > 60 years -16.74 -28.26 -5.23 4.56E-03 ***

Tumor grade (G2 as Ref.)

 G1 34.28 20.08 48.47 3.56E-06 ***

 G3 23.49 9.06 37.92 1.54E-03 **

 G4 22.53 10.51 34.55 2.78E-04 ***

a Multivariate linear regression was used to study the association of epigenetic age acceleration with SDM, adjusting 

age and tumor grade. * p < 0.05, ** p < 0.01, *** p < 0.001

2) The positive association of epigenetic age acceleration with patient overall survival

 We are unable to validate this association, which may be due to data heterogeneity and small sample sizes. The validation data are from multiple studies, which may bring bias for specific population and more heterogenous. In addition, the validation data have small sample size (only 105 samples are eligible for analysis), which reduces the power to detect the difference.

 However, in stratified analyses, we observed the trend that epigenetic age acceleration is positively correlated with patient overall survival. In the younger patients, the hazard ratio is 0.93 (95% CI: 0.84-1.03, p=0.152). Similarly, the hazard ratios in G2, G3 and G4 are 0.77 (95% CI: 0.47-1.28, p=0.317), 0.87 (95% CI: 0.71-1.11, p=0.299) and 0.97 (95% CI: 0.85-1.10, p=0.640) respectively (S5 Table and Figure 4). For the hazard ratios in different molecular subtypes, epigenetic age acceleration shows marginally positive association with patient survival in Mesenchymal-like subtype (HR: 0.85, 95% CI: 0.70-1.13, p=0.094).

We added the validation dataset information in the “Study population” of “Materials and Methods” section and one supplemental table (S2 Table). We added the corresponding results in the “Results” section and three supplemetal tables (S3-S5 Tables) and one figure (Figure 4).

---

## [Decision Letter · Decision Letter 1]

29 Jun 2020

Epigenetic age acceleration and clinical outcomes in gliomas

PONE-D-20-06616R1

Dear Dr. Xu,

We’re pleased to inform you that your manuscript has been judged scientifically suitable for publication and will be formally accepted for publication once it meets all outstanding technical requirements.

Kind regards,

Hiromu Suzuki, M.D., Ph.D.

Academic Editor

PLOS ONE

Additional Editor Comments (optional):

Reviewers' comments:

Reviewer's Responses to Questions

**Comments to the Author**

1. If the authors have adequately addressed your comments raised in a previous round of review and you feel that this manuscript is now acceptable for publication, you may indicate that here to bypass the “Comments to the Author” section, enter your conflict of interest statement in the “Confidential to Editor” section, and submit your "Accept" recommendation.

Reviewer #1: All comments have been addressed

Reviewer #2: All comments have been addressed

2. Is the manuscript technically sound, and do the data support the conclusions?

Reviewer #1: Yes

Reviewer #2: Yes

3. Has the statistical analysis been performed appropriately and rigorously? 

Reviewer #1: Yes

Reviewer #2: Yes

4. Have the authors made all data underlying the findings in their manuscript fully available?

Reviewer #1: Yes

Reviewer #2: Yes

5. Is the manuscript presented in an intelligible fashion and written in standard English?

Reviewer #1: Yes

Reviewer #2: Yes

6. Review Comments to the Author

Reviewer #1: This manuscript is improved after a revision by the authors, and now is suitable for acceptance for PLoS One.

Reviewer #2: The manuscript is well written, and authors have adequately addressed all my concern. I believe this article is suitable to publish in PLOS ONE.

7. PLOS authors have the option to publish the peer review history of their article (what does this mean?). If published, this will include your full peer review and any attached files.

Reviewer #1: No

Reviewer #2: **Yes: **Chiung-Hui Liu

---

## [Editor Report · Acceptance letter]

6 Jul 2020

PONE-D-20-06616R1 

Epigenetic age acceleration and clinical outcomes in gliomas 

Dear Dr. Xu:

I'm pleased to inform you that your manuscript has been deemed suitable for publication in PLOS ONE. Congratulations! Your manuscript is now with our production department. 

Kind regards, 

on behalf of

Dr. Hiromu Suzuki 

Academic Editor

PLOS ONE